# Imaging the Vulnerable Carotid Plaque with CT: Caveats to Consider. Comment on Wang et al. Identification Markers of Carotid Vulnerable Plaques: An Update. *Biomolecules* 2022, *12*, 1192

**DOI:** 10.3390/biom13020397

**Published:** 2023-02-20

**Authors:** David C. Rotzinger, Salah D. Qanadli, Guillaume Fahrni

**Affiliations:** 1Department of Diagnostic and Interventional Radiology, Lausanne University Hospital and University of Lausanne, Rue du Bugnon 46, 1011 Lausanne, Switzerland; 2Faculty of Biology and Medicine, University of Lausanne, 1015 Lausanne, Switzerland; 3Riviera-Chablais Hospital, 1847 Rennaz, Switzerland

We read with great interest the review by Wang et al. entitled “Identification Markers of Carotid Vulnerable Plaques: An Update”, recently published in *Biomolecules* [1]. We congratulate the authors for their comprehensive discussion of the state-of-the-art regarding carotid plaque vulnerability. They approached histological and biochemical aspects, and non-invasive assessment by various imaging methods was covered. Given cerebrovascular disease’s burden on global healthcare, being the second cause of mortality, second only to ischemic heart disease according to the World Health Organization Global Health Estimates [2], being able to understand and detect carotid plaque vulnerability is key to supporting decision-making in clinical practice.

Although we acknowledge that the review is broad, we would like to comment on the strengths and limitations yielded by cervical computed tomography (CT), as the available evidence may need a more tempered interpretation as to its clinical impact. Here, we will focus on selected aspects, including intraplaque hemorrhage (IPH), spectral CT, and plaque calcification. However, we acknowledge that there are many more vulnerability features, most of which have been discussed by Wang et al.

## 1. Hounsfield Unit Thresholds

The authors describe CT numbers of intraplaque material to help better characterize non-calcified components. While CT is a quantitative imaging modality and can therefore help standardize diagnostic criteria, the <25 Hounsfield units (HU) threshold proposed to discriminate IPH is based on a single retrospective study by Saba et al. [3] and is still vastly debated in the literature [4]. As the authors themselves stated [3], their study was preliminary, and the results are yet to be duplicated with different CT systems, acquisition protocols, radiology, and pathology teams. Distinguishing IPH from other non-calcified plaque subtypes, especially lipid-rich non-calcified (LRNC) components on CT, is a non-trivial challenge because both IPH and LRNC have low, overlapping CT numbers < 60 HU. From there, attempting to discriminate IPH—which is considered to have CT numbers lower than 25 HU—seems a bold target, given the numerous factors that can influence attenuation measurements, especially in small regions of interest. Partial volume averaging, tube potential, the density of intraluminal contrast, slice thickness, and reconstruction filter can significantly influence CT numbers, and using absolute thresholds to characterize plaque may be problematic [5,6,7,8,9]. Additionally, CT numbers of blood products range from 40 to about 60 HU, 40 HU being the expected value for circulating blood and 60 HU for clotted blood [10]. Consequently, the underlying rationale for hemorrhage lowering the HU value of non-calcified plaque to under 25 HU remains unsolved.

Notably, other groups could not replicate the findings by Saba et al. For example, Yang et al. performed another retrospective study assessing carotid plaque with histopathological analysis and found IPH to have a median [IQR] CT number of 52 [43–65] HU when associated with plaque ulceration and 57 [44–66] HU when not associated with ulceration, about twice the advocated threshold of < 25 HU [11]. In a smaller cohort, Li et al. found lower CT numbers for LRNC (37.1 ± 15.1 HU) than for IPH (58.4 ± 21.6 HU), so precisely the opposite of what Saba et al. suggested, again seriously undermining the <25 HU threshold’s validity and reproducibility to determine IPH [12]. Likewise, using T1-weighted MRI as the reference standard, U-King et al. measured 47 ± 15 HU in plaques with IPH and 43 ± 14 HU in plaques without [13]. Furthermore, in an imaging study without histopathological analysis, Saba et al. showed that plaques with CT numbers < 25 HU are associated with stroke, but without presenting evidence that IPH actually occurred in those plaques [14]. This is in line with studies pointing out the significant overlap of IPH and LRNC CT numbers [13,15]. Reasons for the discrepancy are currently unclear but may include differences in study population characteristics, different handling of samples, or the fact that various stages of IPH might exhibit variable attenuation. The endpoint might be that low HU values are indicative of high-risk plaque in general, including LRNC, rather than a specific surrogate marker of IPH. As a consequence, the statement that “The combination of the modern MDCTA and analysis software can (…) accurately differentiate IPH and LRNC, and avoid radiation and side effects related to contrast agents” [1] does not hold true and calls for nuance.

## 2. Role of Spectral CT

To further expand the CT-related discussion presented by Wang et al., we want to mention a relatively new technique: spectral CT. Contrary to conventional multidetector CT, which integrates the sum of energy reaching the detector during a projection, spectral CT can discriminate X-ray energy levels to a certain extent. Systems capable of separating two spectral strata were introduced about 15 years ago under the term “dual-energy CT” (DECT) and have since shown to be helpful both in increasing patient safety and diagnostic performance. For carotid plaque characterization, DECT-derived virtual non-contrast can help determine plaque ulceration, as shown by Yuenyongsinchai K. et al. [16]. Interestingly, the same study found IPH to be predictive of 30-day stroke recurrence using a 150 HU cutoff. DECT also proved helpful in mitigating calcium blooming artifacts [17,18,19], differentiating LRNC from other non-calcified plaque components [12], selecting the antiplatelet regimen following acute carotid stenting in tandem occlusions [20], and improving image quality over conventional CT angiography [21,22]. Finally, photon counting CT, the most advanced implementation of spectral CT, by providing improved spatial resolution, better spectral separation, and lower noise, is showing promise in determining fibrous cap and non-calcified plaque components [23,24].

## 3. Calcification as a Vulnerability Marker

Another critical topic mentioned in the review but deserving further discussion is carotid plaque calcification. On several occasions, including in the MRI and CT imaging chapters, the authors mention calcification as a feature of vulnerable plaque. This can be true but also misleading. Our understanding of calcium accumulation along the atherosclerosis progression, involving different inflammatory stages and eventually healing, has recently made notable progress for atherosclerotic carotid disease and coronary artery disease [25]. Calcium build-up occurs due to inflammation-dependent mechanisms involving macrophages that lead to calcium deposition within the plaque’s necrotic core, leading to microcalcifications that are believed to be associated with plaque vulnerability, but are usually not detectable with non-invasive imaging, e.g., CT [26]. ^18^F-Sodium Fluoride (^18^F-NaF) imaging is an exception to the latter statement, and recent research has demonstrated that ^18^F-NaF can specifically detect microcalcification with a resolution beyond that of CT [27,28]. Observations have shown that these microcalcification foci often amalgamate into larger calcium chunks, eventually becoming detectable on CT and MRI, and visible as dot-shaped calcium deposits often referred to as “spotty calcification”. The latter has proven predictive value for the occurrence of ischemic stroke [26,29]. As the inflammation resolves, more calcium is deposited in the healing process, and the plaque undergoes remodeling. At this stage, the term “macrocalcification” is usually used, as deposits can become the dominant plaque component, mixed plaque eventually progressing to calcified plaque, and is supposed to be stabilizing the plaque. Unlike LRNC, large calcification is considered a plaque stability maker [30,31]. Consequently, while small foci of calcium within carotid plaque represent a sign of vulnerability, massively calcified plaque is more likely to indicate healed, stable plaque. Additionally, both calcium density and distribution within the plaque play a role. In a DECT study with a 12-month follow-up, higher plaque density (calcium concentration measured with material decomposition in g/L) was also associated with more stable plaque [32]. Other investigations, including patients with recent neurovascular symptoms [33] and endarterectomy specimens [34], looked into calcium distribution and size. The former concluded that plaques with IPH more frequently exhibited multiple calcifications, particularly when located superficially relative to the lumen. The latter looked into the number and location of calcium foci within the plaque and found that multiple calcifications were associated with both IPH and LRNC. In contrast, basal calcification acted as a protective factor for IPH. On the other hand, some groups found that a high level of carotid artery calcification is not necessarily indicative of lower vulnerability, indicating that plaque vulnerability related to calcium build-up is still insufficiently understood [35].

## 4. Conclusions

When imaging carotid arteries with CT, scrutinizing plaques in search of vulnerability criteria can add incremental value in patient assessment; however, determining plaque components and vulnerability features solely based on attenuation values can be misleading. The absolute < 25 HU threshold advocated in the literature to determine IPH lacks validation. There is contradictory evidence that IPH presents with CT numbers about twice as high, up to 58 HU, highlighting an urgent need for further research aiming to distinguish non-calcified plaque components with CT. Furthermore, the mere presence of calcium in atherosclerotic plaque does not necessarily imply associated inflammation and vulnerability. Here, it is critical to distinguish small, multiple calcium foci within the non-calcified plaque that indicate vulnerability from extended calcification—especially when dense and in basal distribution—that suggests the opposite, namely healed, stable plaque.

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
