# Peer review of "Imaging the Vulnerable Carotid Plaque with CT: Caveats to Consider. Comment on Wang et al. Identification Markers of Carotid Vulnerable Plaques: An Update. Biomolecules 2022, 12, 1192"

_biomolecules, 2023, doi:10.3390/biom13020397_

Round 1

Reviewer 1 Report

The comments is in the file.

Author Response

We thank the reviewer for the comments. As mentioned in the review, our manuscript is a short commentary highlighting key concepts regarding two selected CT features: IPH and calcification. We acknowledge that there may be many more which are not covered because then our work would become a full-length review; we modified our text in that regard. Additionally, we added a comment about the potential of photon-counting CT to determine fibrous cap.

Reviewer 2 Report

It was a well-written letter focusing on the possible concerns on a recently published review article. The authors proposed several concerning points on CTA assessments of carotid plaques. 

The authors listed results of several studies to support their points. The overlaps of CTA values of lipid, IPH, and fibrous components of carotid plaques have been reported. And the differentiation of different non-calcified components based on CT attenuations should be verified in the further studies.

I have several concerns and suggestions.

1.     As for the IPH detection, I agree with the views of the authors. However, there were some controversial results in previous studies. The reasons should be discussed briefly. Was it due to the included subjects or samples? Or the CTA technique or parameters? the CT attenuation might be related to the different stages of the IPH? Or other reasons? It might be helpful for the future study.

2.     In the review article, the dual-energy CT applied in recent years was totally ignored. Only the MDCT was mentioned, which was not a new technique. I suggest the authors add the contents of the performance of DECTA (DLCTA) in the assessment of the carotid plaques. It would fill in the omissions of the review article and give a more comprehensive understanding of CTA technique.

3.     As for the calcification, I agree with the authors that it is not right to simply classify the calcification as one of the vulnerable features. (“High-resolution MRI can identify several classical features of vulnerable plaques, such as calcification, IPH, inflammatory tissues, thin and ruptured caps and LRNC [138,139]. Accordingly, the identified features may provide guidance for risk stratification of carotid plaques [140]. --Identification Markers of Carotid Vulnerable Plaques: An Update). 

Calcification is considered as a stable sign for most cases. However, there were some studies in recent years which demonstrated the surface/mixed calcification protruded into lumen might be the feature of vulnerable plaques. I don’t think it is accurate to state that the macro-calcification is stable. It might be related to the location and the morphology of the calcification in the plaque. I think it would be better to include the related refs (1-4) and carefully summary the current results of previous studies.

1) Fan Z.X., Yuan S.J., Li X.Q., Yang T.T., Niu T.T., Ma L., et al. Preliminary study on the differentiation of vulnerable carotid plaques via analysis of calcium content and spectral curve slope by using gemstone spectral imaging. Exp Ther Med. (2022). 23:325 DOI: 10.3892/etm.2022.11254.

2) Lin R., Chen S., Liu G., Xue Y., and Zhao X. Association Between Carotid Atherosclerotic Plaque Calcification and Intraplaque Hemorrhage: A Magnetic Resonance Imaging Study. Arterioscler Thromb Vasc Biol. (2017). 37:1228-1233 DOI: 10.1161/atvbaha.116.308360.

3) Pini R., Faggioli G., Fittipaldi S., Vasuri F., Longhi M., Gallitto E., et al. Relationship between Calcification and Vulnerability of the Carotid Plaques. Ann Vasc Surg. (2017). 44:336-342 DOI: 10.1016/j.avsg.2017.04.017.

4) Xu X., Hua Y., Liu B., Zhou F., Wang L., and Hou W. Correlation Between Calcification Characteristics of Carotid Atherosclerotic Plaque and Plaque Vulnerability. Ther Clin Risk Manag. (2021). 17:679-690 DOI: 10.2147/tcrm.S303485.

5. By the way, I also think it is not accurate to say that “The combination of the modern MDCTA and analysis software can … accurately differentiate IPH and LRNC, and avoid radiation and side effects related to contrast agents…” (--Identification Markers of Carotid Vulnerable Plaques: An Update).

6. Considering the points proposed in this letter, I suggest the authors may remind the readers in the conclusion that carotid plaques components and vulnerability analysis based only on attenuations should be explained very carefully. And the differentiation of different non-calcified components based on CT attenuations should be verified in the further studies.

Author Response

1. We thank the reviewer for this helpful suggestion. We revised and added the missing points.

2. We agree with the reviewer and had considered discussing spectral CT, finally deciding not to because the available data did not truly help resolving the CT number question. But we are convinced that spectral CT will bring new insights and since the reviewer advises us to comment DECT, we are happy to add a paragraph on that topic.

3. 4. We appreciate this thought-generating comment and revised the paragraph dealing with calcification, and the conclusion. We have also included the 4 suggested references.

5. We the reviewer for picking up this particular statement. We have added a comment to highlight this issue.

6. We have improved the conclusion with this valuable suggestion.

Reviewer 3 Report

Dear Authors and Editor,

This comment article points at important shortcomings in the recently published review article  "Identification Markers of Carotid Vulnerable Plaques: An Update.” The comment article is highly interesting and clarifies important points in our understanding of plaque vulnerability and what shortcomings imaging modalities have when the task is to characterize tissues with very small volume.

The authors address two specific points, a CT number threshold < 25 Hounsfield units (HU) proposed to discriminate intraplaque haemorrhage and the role of carotid plaque calcification in plaque vulnerability. Both points are thoroughly discussed in the comment using relevant references and demonstrate that the review article refers to and summarizes the literature about these two topics wrong.

In the 5th paragraph line 71 (about microcalcification) the authors state: “but usually not detectable with non-invasive imaging, e.g., CT”.  In my opinion the authors here should add that the literature covering 18F-Sodium Fluoride (18F-NaF) for imaging microcalcification activity in atherosclerotic plaque progression has been extensive the last years. Another shortcoming with the review article is that the chapter 3.3. Positron Emission Tomography (PET) is written without an understanding of the modality and a substantial amount of publications are lacking – but that is not my task to comment on.

Author Response

We thank the reviewer for the positive feedback and helpful comment. We have revised the 5th (which is now the 6th paragraph following major revisions) with this interesting fact about 18F-NaF imaging. For the rest, the reviewer certainly has a good point the PET imaging literature review could be improved, but we refrained to comment on this topic since we focus on CT angiography.